# A New Generation of Postbiotics for Skin and Scalp: In Situ Production of Lipid Metabolites by *Malassezia*

**DOI:** 10.3390/microorganisms12081711

**Published:** 2024-08-19

**Authors:** Martin Patrick Pagac, Mathias Gempeler, Remo Campiche

**Affiliations:** DSM-Firmenich, Perfumery & Beauty, Wurmisweg 576, CH-4303 Kaiseraugst, Switzerland; mathias.gempeler@dsm-firmenich.com (M.G.); remo.campiche@dsm-firmenich.com (R.C.)

**Keywords:** skin microbiome, *Malassezia*, lipid mediator, oxylipin, prebiotics, postbiotics, inflammation, cosmetics

## Abstract

Effects of pre- and probiotics on intestinal health are well researched and microbiome-targeting solutions are commercially available. Even though a trend to appreciate the presence of certain microbes on the skin is seeing an increase in momentum, our understanding is limited as to whether the utilization of skin-resident microbes for beneficial effects holds the same potential as the targeted manipulation of the gut microflora. Here, we present a selection of molecular mechanisms of cross-communication between human skin and the skin microbial community and the impact of these interactions on the host’s cutaneous health with implications for the development of skin cosmetic and therapeutic solutions. *Malassezia* yeasts, as the main fungal representatives of the skin microfloral community, interact with the human host skin via lipid mediators, of which several are characterized by exhibiting potent anti-inflammatory activities. This review therefore puts a spotlight on *Malassezia* and provides a comprehensive overview of the current state of knowledge about these fungal-derived lipid mediators and their capability to reduce aesthetical and sensory burdens, such as redness and itching, commonly associated with inflammatory skin conditions. Finally, several examples of current skin microbiome-based interventions for cosmetic solutions are discussed, and models are presented for the use of skin-resident microbes as endogenous bio-manufacturing platforms for the in situ supplementation of the skin with beneficial metabolites.

## 1. Introduction

The skin, as the largest organ of our body, serves as a natural barrier against the outside environment. Providing a heterogeneous surface with distinct dry, moist, or oily microenvironments, the skin can therefore be regarded as a diverse ecosystem. The skin structure is composed of different specialized cell types, such as keratinocytes, melanocytes, dendritic cells, and fibroblasts, and possesses appendages like the hair follicles and sebaceous as well as sweat glands. For that reason, the skin offers a diverse and rich environment for population by a myriad of microorganisms, each evolutionary adapted to inhabit one of the many skin niches [1]. Traditionally, skin is divided into three primary layers (the dermis, epidermis, and subcutaneous adipose tissue), which each contribute specialized functions to human health. However, given the importance of host–microbe interactions for skin homeostasis, the task of a fourth layer has now been attributed to the skin microbiota. Recently, it has become evident that the skin microflora, defined by the entire skin-resident bacterial, archaeal, fungal, and arthropodal organisms, including viruses, contributes actively to skin homeostasis and health [2]. While under unfavorable circumstances, some skin microbes can cause harm to their hosts, the presence of other microbial species can be beneficial for the skin by providing protection against the invasion of pathogens and boosting the cutaneous immune response.

Skin homeostasis is impacted by a complex interplay between inherited factors, such as the consequences of aging-associated structural and functional changes to the skin [3], and environmental factors, such as exposure to pollution [4,5], solar UV irradiation [6], and the composition of the microbial flora [7]. Some of these combinatory intrinsic and extrinsic factors may eventually trigger skin inflammation and consequently impair epidermal barrier function and cause irritating skin symptoms such as pruritus, an itchy and painful skin sensation [8,9]. In fact, the number of people claiming to experience sensitive skin, a clinical condition defined by the presence of different unpleasant and painful sensory perceptions, is steadily increasing [10]. Traditionally, commercial skincare products aim to reduce symptoms of inflammation by using soothing, cooling, moisturizing, or hydrating ingredients. Commonly added anti-inflammatory compounds that fight the cause and initiation of inflammatory signs include, but are not limited to flavonoids, zinc preparations, polyphenols, carotenoids [11], as well as niacinamide [12], vitamin E, and certain fatty acids [13]. In contrast to anti-inflammatory metabolites, pro-resolution programs are activated and stimulated by pro-resolving compounds, which ultimately lead to the termination of an established inflammation [14] by inhibiting neutrophil migration and enhancing macrophage phagocytosis [15]. The concept of incorporating such pro-resolving mediators as bioactives into topical products to reduce symptoms associated with existing skin inflammations at a cell-molecular level holds high promise for the development of next-generation cosmetic or therapeutic solutions. Intriguingly, common fungal components of the human skin and scalp microflora, such as yeasts belonging to the *Malassezia* genus, were recently shown to produce lipid metabolites with potent anti-inflammatory and pro-resolving properties [16,17] and to positively correlate with skin surface eicosanoids [18]. Some of these lipid metabolites belong to a diverse lipid class of oxygenated, polyunsaturated fatty acids (PUFA), and are a central interactive tool to facilitate communication between host and microbe [19]. The exact physiological functions of these microbial-derived lipid metabolites, however, are not yet fully understood.

In this article, we discuss and evaluate the cosmetic and therapeutic potential of common skin- and scalp-resident microbes, such as *Malassezia* yeasts, to be used as endogenous bio-manufacturing platforms for the in situ production and consequent supplementation of skin with beneficial pro-resolving and anti-inflammatory lipid metabolites with soothing properties.

## 2. Lipid-Mediated Communication between the Skin and Its Microflora

Communication is key for purposeful interactions between different organisms. While humans have developed sophisticated verbal and non-verbal tools for communication with each other, lower organisms evolved with alternative molecular languages for inter- and intra-species cross-communication, even allowing interactions between different kingdoms of life. Classical examples of such cross-kingdom communications are between fungi and plants and involve lipid mediators as language tools to either increase fungal pathogenicity or foster symbiotic relationships [19]. The lipid mediators used in the plant world belong to the phyto-oxylipin family and include jasmonates and their volatile and diversely conjugated derivatives [20]. The term “oxylipin” refers to oxygenated compounds metabolized from fatty acids through at least one mono- or dioxygenase oxygenation reaction [21], and is now used to describe a wide variety of bioactive lipid mediators [22]. Several microbial-derived lipid mediators have also been shown to interact with and exert biological activity on mammalian cells. Predominantly fungal pathogens, such as *Candida albicans* and *Aspergillus fumigatus*, produce lipid mediators with structures similar or identical to the ones synthesized by humans. The human host cells, such as epithelial or endothelial cells, macrophages, and neutrophils, recognize these fungal-derived molecules and respond with altered biological functions, ranging from decreased cytokine production, inhibited phagocytosis, and anti-fungal activity to a reduced survival rate [23].

Recently, researchers reasoned that microbes commonly found on the skin could use similar mechanisms to interact with the human host’s skin. Indeed, targeted lipidomic analysis revealed that several in vitro cultured skin-resident *Malassezia* species, such as *Malassezia globosa*, *Malassezia restricta*, *Malassezia sympodialis,* and *Malassezia furfur*, were able to produce bioactive lipid mediators that could also be detected on the surface of human skin [16]. These findings suggest that some of these lipid mediators are either directly produced by *Malassezia* on the surface of human skin and in hair follicles or are synthesized via metabolic pathways shared between human skin cells and the skin mycobiome [17]. Indeed, microbes derived from murine gut were shown to metabolize the bioactive lipid mediator 9,10-DiHOME directly from linolenic acid (LA), thereby inducing regulatory T cell (Treg) differentiation and maintaining intestinal immune homeostasis [24]. Of important note, by no means is the host–microbe communication lexicon limited to lipids and their derivatives, as conversation tools also include a plethora of other small metabolites, mycotoxins [25], peptides, and proteases to modulate the host’s immune response [1] and direct quorum sensing mechanisms [26].

In molecular biology, all physical interactions between molecules that take place in a cellular unit are traditionally described as the interactome [27]. Recently, Khmaladze and colleagues widened the terminological boundaries by introducing the “skin interactome model”, describing the complex mechanisms by which combinations of different factors impact cutaneous health [28]. While genetic, exposomal, and microbial factors individually have a significant effect on skin homeostasis, under physiological conditions it is rather the mutual and interactive influence that plays a role in skin health maintenance. For example, several intrinsic and extrinsic factors, such as age, menopausal status, pollution, sun exposure, and cosmetic product usage, have been shown to impact skin microbiome compositions: menopausal status [29] and aging-associated [30] changes in skin physiological parameters significantly affected skin microbiome profiles and chronic exposure levels of polycyclic aromatic hydrocarbon (PAH) pollutants correlated with changes in composition and functional capacities of the cutaneous microbiota [31]. Furthermore, skin microbiome compositions and the re-colonization potentials of skin-resident microbes were profoundly affected by ultraviolet radiation (UVA and UVB) [32], whereas the application of sunscreens preserved the skin microbiome upon erythemal UV exposure [33]. Lastly, topical application of synthetic skin care products reduced skin microbiome diversity, a feature that is considered to be associated with skin health [34], and commonly used cosmetic preservatives impacted skin-resident microflora dynamics in vitro [35]. Not surprisingly, the importance of microbiome-friendly formulations for skin integrity has widely been accepted by the cosmetic industry, and experimental assessments of microbiome friendliness allowing validation and acceptance of associated claims have been proposed [36].

This raises the interesting question of whether a concomitant dysbiotic skin microflora leads to an unbalanced bioactive metabolite profile and consequently adversely affects the health status of the skin. After all, a better understanding of how skin interactome-derived metabolites are produced will be essential to further develop a strategy to preserve good cutaneous health conditions. The concept of influencing skin homeostasis through modulation of the skin interactome network and consequent production of bioactive metabolites can be translated to a diverse range of cosmetic and therapeutic applications. To decomplex the substantiation of the significance of host–microbe interactions for skin health, the focus of this review will be on *Malassezia*-derived lipid mediators. 

## 3. *Malassezia*, the Most Predominant Fungal Genus on Healthy Human Skin and Scalp, Produces Bioactive Lipid Mediators

*Malassezia* yeasts are a genus of fungi and commonly inhabit human and animal skin. Even though the numerical abundance of the bacterial kingdom clearly takes the lead on the skin, the biomass of fungi, collectively termed the mycobiome, is proportionally much higher with respect to cellular size/genomic material ratios: the average diameter of *Malassezia* cells is 10 times larger than that of bacteria, and, as such, the cellular volume, and, consequently, the metabolically active biomass, is 100-fold greater [37]. Inequitably, the skin mycobiome, specifically its most prominent *Malassezia* member, was long understudied and its impact on cutaneous health underestimated [38]. Usually, *Malassezia* is well tolerated by the immune system of human skin. But occasionally, these yeasts turn pathogenic and are associated with skin disorders, such as pityriasis versicolor, seborrheic dermatitis/dandruff, atopic eczema, atopic dermatitis (AD), and psoriasis [39]. Outside a dermatological context, proximal interactions between the normally skin-resident fungus *Malassezia* and other cellular niches exist, thereby accelerating, for example, pancreatic cancer progression [40]. *Malassezia* was also found to be present in gastrointestinal and lung tumors and significantly reduced survival rates in breast cancer [41,42]. Furthermore, several *Malassezia* species are associated with Parkinson’s disease [43] and inflammatory bowel disease (Crohn’s disease) [44]. Lastly, a recent population-based study revealed that the intestinal presence of *Malassezia restricta* was associated with the promotion of alcohol-induced liver injury [45]. Conversely, *Malassezia* has also been shown to exert mutualistic and preventative activity in human and animal skin [46]; for example, by suppressing invasion of the skin by the fungal pathogen *Candida auris* [47]. Hence, a better understanding of the role of *Malassezia* in these diseases may help to design targeted interventions. 

Even though a causative link between *Malassezia* and skin diseases has yet to be demonstrated, several hypotheses exist: a recent study showed that *Malassezia restricta* mediates the production of squalene mono-hydroperoxidation and malondialdehyde. These sebum peroxidation products act on the epidermis and alter skin barrier functions, thereby potentially triggering and maintaining dandruff. Blocking the fungus-derived lipoperoxide production is hypothesized to lead the way for a novel therapeutic anti-dandruff treatment strategy [48]. A more traditional hypothesis holds the lipophilicity and lipo-dependency of *Malassezia* responsible for the development of dandruff: most *Malassezia* species require an exogenous source of fatty acids (FA) for survival, as over the course of evolution they lost the genetic capability to de novo produce FA and metabolize carbohydrates. This absolute requirement for lipids partly explains why *Malassezia* is predominantly found on oily skin regions with high sebaceous gland activity, such as the forehead, cheeks, scalp, chest, and upper back [49]. The *Malassezia* genomes, as well as the lipophilic *Cutibacterium acnes* species, contain a plethora of genes that encode lipases that are potentially secreted onto skin [50,51], where they release saturated and polyunsaturated fatty acids (PUFA) from sebum lipids that are produced by host epidermal sebaceous glands. Due to a lack of Δ^9^-desaturase activity [52], of these, only the saturated FA are consumed by *Malassezia*, whereas the unsaturated FA, such as oleic acid, are left behind and are believed to induce the hyperproliferation of epidermal keratinocytes and consequently cause dandruff-like flaking of human scalp skin [53]. Similar to microbial liberation of FA from sebum, solar UV-B irradiation of skin induces activation of phospholipase A2, resulting in the hydrolysis of FA, predominantly the PUFA arachidonic acid from cellular membranes [54]. Several studies have revealed that dandruff is associated with bacterial and fungal dysbiosis on the scalp. For example, the alpha diversity (Shannon diversity index) of the fungal population as well as the *M. restricta* to *M. globosa* ratio was significantly higher on dandruff-affected scalp skin compared to the healthy scalp [55]. The inhibition of lipase activity from *M. restricta* is hence considered a potential target for dandruff therapy [56]. More drastic approaches include the treatment of dandruff with shampoos and rinse-off products containing anti-fungal actives, such as zinc pyrithione (ZPT), leading to a reduction of yeast cells on the scalp and an associated reduction in flaking [57].

Microbiome-driven hydrolysis of FA, particularly long-chain PUFA, together with de novo synthesized PUFA and PUFA obtained from dietary intake, can ultimately serve as precursors for the production of lipid mediators, a diverse class of lipids with highly potent biological activities. Specifically, lipid mediators are generated either through spontaneous non-enzymatic free-radical-catalyzed or highly regulated enzymatic oxygenation of omega-3 (n-3) or omega-6 (n-6) PUFA. The enzymatic reactions are facilitated mainly by lipoxygenases (LOX), cyclooxygenases (COX), or via pathways involving epoxygenases of the cytochrome P450 family [58]. 

Lipid mediators are characterized by being involved in interactions between the host and microbes and having strong immuno-modulatory functions [19]. Lipid mediators that emerged from n-3 PUFA have anti-inflammatory or pro-resolving properties. In agreement with several study outcomes, lipid mediators of n-6 origin are implicated mostly in pro- and, to a lesser extent, anti-inflammatory pathways [59]. Immunomodulatory action is context-dependent: despite PGE_2_ being a classic pro-inflammatory lipid mediator representative, it also has potent anti-inflammatory properties [60]. The detailed mechanistic roles of pro-resolving and anti-inflammatory lipid mediators in terminating acute inflammations are comprehensively discussed elsewhere [14].

Intriguingly, as stated above, multiple in vitro cultured *Malassezia* species were shown to produce the same lipid mediators that could also be found on human skin [16]. The list included, amongst others, the n-6 linoleic acid-derived 9-HODE, 9,10-DiHOME, and 12,13-DiHOME lipid mediators with well-characterized pro-inflammatory properties that were also shown to be elevated in psoriatic skin [61]. This suggests that these *Malassezia*-derived lipid mediators are involved in the development and progression of skin disorders caused by cutaneous dysbiosis. On the other hand, the cultured *Malassezia* fungus is also capable of producing a series of anti-inflammatory lipid mediators, as well as Maresin-1 [17], with potent bioactive properties involved in promoting the resolution of the inflammatory response (see Table 1 below). N-3 PUFA are competitive substrates for the enzymes and products of n-6 PUFA metabolism [62]. Excess n-3 substrate availability could hence suppress *Malassezia*-derived production of pro-inflammatory lipid mediators and contribute to the resolution of inflammatory conditions at affected skin sites (see Figure 1). Indeed, oral supplementation of n-3-enriched fish oil to healthy individuals affected the epidermal lipidome by depleting n-6-derived lipid mediators and inhibiting the migration of the Langerhans cells [63] that form a dense network of immune system sentinels in the epidermis [64].

Moreover, a recent lipidomic analysis of a subset of *Malassezia* species revealed the presence of bioactive FA esters of hydroxyl FAs (FAHFAs) with attributed antidiabetic and anti-inflammatory properties [65,66], thus expanding the fungal repertoire of immune-modulatory compounds beyond the here discussed classical lipid mediators.

A detailed exploration and characterization of genetic pathways leading to the production and secretion of these highly potent lipid mediators will ultimately facilitate the discovery of novel therapeutic targets for the treatment of inflammatory skin diseases and the development of microbiome-based cosmetic solutions.

**Table 1 microorganisms-12-01711-t001:** List of 23 anti-inflammatory and pro-resolving lipid mediators produced by *Malassezia* [16,17]. PUFA precursors and relevant publications are indicated for each lipid mediator species. Lipid mediators are grouped by relation. Abbreviations for listed precursor PUFA: Docosahexaenoic acid (DHA), Linoleic acid (LA), α-linolenic acid (ALA), Eicosapentaenoic acid (EPA), Dihomo-γ-linolenic acid (DGLA), Arachidonic acid (ARA).

Lipid Mediator Species	Immunomodulatory Effects	Precursor
Maresin-1 (MaR1)	Anti-inflammatory action in psoriasis through inhibition of IL-17A production [67,68].	DHA (22:6 n-3)
13-HODE, 13-oxoODE	Anti-inflammatory [69], by maintaining normal keratinocyte proliferation [70].	LA (18:2 n-6)
8-HETrE, 15-HETrE	Anti-inflammatory effect by antagonizing the synthesis of ARA-derived pro-inflammatory mediators [71,72].	DGLA (20:3 n-3)
14,15-DHET	In vitro, 14,15-DHET impaired neutrophil chemotaxis, acidification, CXCR1/CXCR2 expression, and reactive oxygen species (ROS) production [73].	ARA (20:4 n-6)
5,6-diHETrE, 8,9-diHETrE, 11,12-diHETrE, 14,15-diHETrE	Anti-inflammatory effects by activating the peroxisome proliferator-activated receptor alpha (PPARα) pathway [74].	ARA (20:4 n-6)
5,6-EET; 8,9-EET; 11,12-EET; 14,15-EET	Improvement of impaired wound healing by altering inflammatory response in wounds [75].	ARA (20:4 n-6)
13-HOTrE	Mediating anti-inflammatory effects by inactivating NLRP3 inflammasome [76].	ALA (18:3 n-3)
4-HDoHE; 10-HDoHE; 11-HDoHE; 16-HDoHE	Pro-resolving and anti-inflammatory action through reduction of LPS-induced pro-inflammatory cytokine mRNA expression [77].	DHA (22:6 n-3)
5-HEPE, 8-HEPE, 15-HEPE	Anti-inflammatory effect, attenuates the biosynthesis of pro-inflammatory eicosanoids by polymorphonuclear neutrophils [78].	EPA (20:5 n-3)
PGE_2_	Negative regulation of inflammation by inhibiting CCL5 and TNF-α expression in activated macrophages [79].	ARA (20:4 n-6)

**Figure 1 microorganisms-12-01711-f001:**
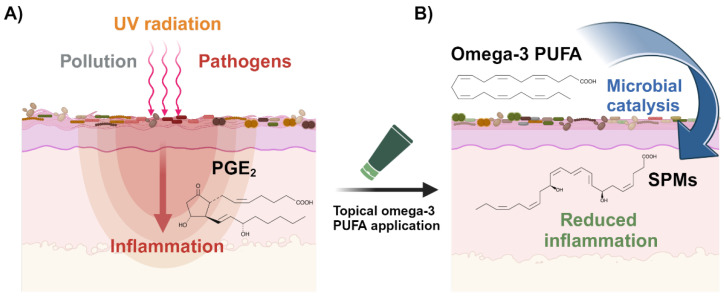
Restoration of skin homeostasis via enzymatic skin microbial conversion of topical omega-3 PUFA into anti-inflammatory/pro-resolving lipid mediators: (**A**) Skin inflammation and the associated burdens, such as redness and itchiness, can have several triggers, including environmental exposures, as well as a dysbiotic skin microflora. As an example, pathogenic *Cutibacterium acnes* or *Staphylococcus aureus* bacterial strains can directly stimulate the skin immune system via the production of immunomodulators, such as, for example, the pro-inflammatory lipid mediator Prostaglandin E2 (PGE_2_; depicted in the scheme) [80]. (**B**) Topical supplementation of skin with excess n-3 PUFA, such as EPA or DHA (DHA structure depicted in the scheme), leads to enzymatic conversion into specialized pro-resolving mediators (SPMs), such as Maresin-1 (MaR1; depicted in the scheme) and/or other anti-inflammatory lipid mediators with postbiotic properties, and reduces products of n-6 PUFA metabolism through direct substrate competition [62]. Enzymatic production of SPMs is mediated by skin-resident fungi (microbial reactions symbolized by blue arrow), such as *Malassezia* yeasts, thereby resolving inflammation and restoring skin homeostasis. Microbial SPM synthesis can additionally be boosted through support of skin microfloral activity by topical supplementation with pre- or probiotics (see Figure 2).

## 4. Harnessing the Power of Skin-Resident Microbes for In Situ Bioprocessing of Anti-Inflammatory and Pro-Resolving Mediators 

The importance of highly bioactive lipid mediators for regulating the cutaneous immune system has long been appreciated and is comprehensively described elsewhere [84]. Briefly, the beneficial effects of n-3 PUFA, the main building blocks for biosynthesis of anti-inflammatory and pro-resolving lipid mediators, range from regulating the development and outcome of several inflammatory skin diseases and maintenance of epithelial barrier integrity to protection against oxidative stress-induced apoptosis [85,86]. Moreover, aging is associated with an increasing chronic inflammatory disease pathogenesis [87], likely due to impaired resolution of inflammation in aged human skin. Thus, lipid mediators in the skin have the potential to reverse inflammatory reactions not only caused by a disbalanced microfloral composition, pollution, UV, and photoaging but also chronological aging. 

From a skincare perspective, the question to be answered is how lipid mediator production on the skin surface could be modulated. Interestingly, as mentioned above, dietary supplementation of n-3-rich fish oil was shown not only to generally impact the cutaneous lipid mediator profile [63] and to ameliorate skin inflammatory processes, thus helping with disease management [88], but also to protect against UVB-induced photodamage of human skin [89] and to promote wound healing in mice [90]. These findings not only place emphasis on the importance of understanding the mechanisms that inter-connect distal organs such as the gut and skin but also raise the question of how nutritional n-3 PUFA are converted into bioactive lipid mediators in human skin. One plausible explanation is that the enzymatic oxygenation of circulation-derived PUFA is mediated by skin-resident microbes, such as *Malassezia* yeasts, which have been shown previously to be capable of doing so (discussed above). Intriguingly, topically applied n-3 PUFA, formulated into cosmetic skincare products, could potentially be directly metabolized in situ by skin-resident microbes into postbiotics, such as lipid mediators with anti-inflammatory and pro-resolving activities (see Figure 1). The definition of a postbiotic has been proposed as a “preparation of inanimate microorganisms and/or their components that confers a health benefit on the host” [91]. Amongst other benefits, several inflammatory cutaneous disorders associated with *Malassezia* could be prevented, relieved, or even resolved as a consequence of enzymatic production of postbiotics by the fungi itself. Furthermore, in contrast to the more stable PUFA [92], lipid mediators are prone to rapid metabolic inactivation and are labile, with very short half-lives in vivo [93]. From a practical point of view, integration of n-3 PUFA substrates into skincare formulations, rather than lipid mediator products that are prone to spontaneous degradation, would be more cost-effective and sustainable. 

Unlike other tissues, the skin lacks delta-5 and delta-6 desaturase activity and is hence not capable of converting LA to ARA or ALA to EPA and DHA [94,95]. Because of the skin’s inability to produce these long-chain metabolites, DHA and EPA are also considered essential nutrients for the skin. Topical application of DHA and EPA hence could accelerate the capability of the skin to reduce or resolve inflammation.

N-3 PUFA are essential dietary components and abundant in algae, the primary food source for fish. Plant-based diets, common in veganism, are low in these elements, particularly DHA and EPA [96]. While a meat-free diet is undoubtedly associated with various health benefits, not limited to reducing the risk of developing chronic diseases such as metabolic syndrome [97], a large US population-based study revealed a significant connection between a vegan diet and increased prevalence of eczema [98], an inflammatory skin condition. While the therapeutic potential of nutritional n-3 PUFA metabolites for inflammatory skin diseases has been appreciated previously [85], scientific evidence for in situ microbial production of soothing metabolites at affected skin areas after direct topical n-3 PUFA application has yet to be provided with a relevant clinical study, and if verified, may prove to be more effective and sustainable. Indeed, the International Scientific Association for Probiotics and Prebiotics (ISAPP) defined PUFA only as candidate prebiotics because of the current lack of scientific evidence of health benefits for the host [99]. 

Besides nutritional consumption of n-3 PUFA supplements, integration of n-3 PUFA into cosmetic products for topical application is seeing commercial potential due to their antioxidant, anti-inflammatory, and antibacterial properties [100,101]. As an example, microalgae are capable of de novo synthesizing both n-3 and n-6 PUFA, and their integration into vegan-friendly cosmetic applications as a postbiotic ingredient, together with a rich accommodating lipid profile, is seeing a trend in the personal skincare cosmetic industry [102]. Nonetheless, oxidated PUFA are characterized by an unpleasant odor, thus limiting marketing strategies. Particularly, EPA- and DHA-rich oils with a high degree of unsaturation are prone to oxidation [103] when exposed to oxygen, heat, and light [104]. Fortunately, efforts to encapsulate PUFA to prevent auto-oxidation or to integrate ingredients to mask the odor were previously successfully undertaken [105].

As represented in Figure 1, the discovery that skin-resident *Malassezia* yeasts can produce lipid mediators, raises the hypothetical question of whether it is not the n-3 PUFA contained within cosmetic products per se, but rather their enzymatically oxygenated derivatives that contribute to the anti-inflammatory and antioxidant effects. Above all, the skin microflora has now been discovered as an additional dimension in directly fine-tuning the delivery of newly in situ synthesized bioactives onto skin and in regulating their degree of potency [7,106].

However, the biosynthetic pathways leading to the production of these lipid mediators in *Malassezia* are not yet characterized [16]. While bacterial lipoxygenases, enzymes that catalyze the deoxygenation of PUFA, have been discovered [107], knowledge about fungal genes required for the expression of genes involved in the metabolism of oxylipins is scarce, possibly due to their diversity and exhibition of unusual catalytic activities [108]. Indeed, homologous sequences with similarities to known pro- and eukaryotic PUFA oxygenases have not been identified yet in the *Malassezia* genomes. We, therefore, hypothesize that the *Malassezia* genes involved in lipid mediator biosynthesis evolved independently, therefore highlighting their biological importance as a conserved biochemical tool for host–microbe interactions. Further research will be needed to reveal the location of lipid mediator production, i.e., whether PUFA substrates are imported to be metabolized by intracellular enzymatic mechanisms or whether said enzymes are secreted into the extracellular matrix to act on skin-associated PUFA. The identification and characterization of microbial enzymes with PUFA oxygenation activities will allow the development of cosmetic and therapeutical agents with either inhibitory or activating properties for the regulation of respective pro- and anti-inflammatory lipid mediator production. Additionally, modulation of the skin microbiome at the taxonomic species level can be used to indirectly control levels of these bioactives as a strategy for personalized microbiome intervention to reduce the burdens associated with inflammatory skin diseases and improve cutaneous homeostasis and beauty. On the conceptual level, rebalancing of the skin microfloral composition through the targeted topical application of selected pre- and probiotics could potentially reduce or enhance microbial metabolite production with harmful or favorable properties, respectively (see Figure 2). As mentioned above, since the conversion of n-3 and n-6 PUFA is governed by the same series of enzymes, competition exists between the n-3 and n-6 PUFA substrates for these enzymes [62]. Hence, topical supplementation of n-3 PUFA is expected to increase the n-3 to n-6 PUFA ratio on the skin surface, leading to reduced synthesis of inflammatory lipid mediators from n-6 PUFA and elevated production of anti-inflammatory and pro-resolving n-3 PUFA metabolites.

Consensus definitions of probiotics are “live microorganisms that, when administered in adequate amounts, confer a health benefit on the host” [107], whereas a prebiotic is a “substrate that is selectively utilized by host microorganisms conferring a health benefit” [108]. A synbiotic is defined as “a mixture comprising live microorganisms and substrate(s) selectively utilized by host microorganisms that confers a health benefit on the host” [109]. 

A balanced skin microflora metabolizes anti-inflammatory lipid mediators, such as 13(S)-HOTrE (structure shown), from topically supplemented n-3 PUFA, such as the candidate prebiotic ALA (structure depicted in the scheme), at the expense of pro-inflammatory lipid mediators. For simplicity reasons, pathways involving synbiotics and postbiotics with prebiotic properties are not included in the scheme.

## 5. Exploring Current Microbiome-Based Interventions for Skin Cosmetic and Therapeutic Applications

The concept of directly incorporating bioactive ingredients, such as endocannabinoids, growth factors, microbiome modulators, and antioxidant enzymes, into skincare products to upregulate epidermal lipid synthesis, improve epidermal barrier function, and decrease itch and inflammation, reflected by enhanced epidermal skin sensation and appearance, was proposed before [109,110]. Even though this has been neglected in the cosmetic skin industry, products do exist that claim to boost the production of pro-resolving lipid mediators in the skin [111]. Recently, Duroux R. et al. detected an increased abundance of pro-resolving mediators in the scalp sebum of volunteers suffering from dandruff following hair treatment with a plant extract derived from *Anetholea anisita* [112]. However, to the best of our knowledge, cosmetic solutions that involve skin microfloral components as central elements for the epidermal in situ bio-manufacturing of anti-inflammatory or pro-resolving lipid mediators are not commercially available. Nevertheless, the potential use of microorganisms in the form of topical probiotics for the production of beneficial postbiotics on the skin has been conceptually explored previously [113]. For example, ammonia-oxidizing bacterial *Nitrosomonas eutropha* strains produce nitric oxide, a molecule with potential anti-microbial and anti-inflammatory properties. Results from clinical trials involving topical application of this probiotic showed an improvement in pruritus [114] as well as reduced appearance of wrinkles [115]. 

Also, Malassezin is a natural indole derivative and aryl hydrocarbon receptor agonist produced by the commensal skin-resident fungus *Malassezia furfur*, which has been causatively associated with the skin depigmentation disorder pityriasis versicolor [116]. Studies have shown that topical application of isolated Malassezin decreases epidermal discoloration, such as photoaging-induced hyperpigmentation [117], possibly due to specific apoptotic effects on primary human melanocytes [118]. As such, Malassezin, a novel agent unique to the skin microflora, is currently available as a commercial ingredient for skin-whitening applications, and ideally only targets selectively hyperpigmented skin while preventing depigmentation of normally pigmented skin. Similarly, the opportunistic skin pathogen *Cutibacterium acnes* produces propionic acid, which may be as well an effective and non-toxic alternative solution for hyperpigmentation [119]. Pityriacitrin is another *Malassezia furfur*-derived indole alkaloid and was shown to possess potent UV protective properties in yeast, bacteria, and humans [120]. The UV-absorbing effects of Pityriacitrin were not only observed to impact several other bacterial and fungal members of the human skin microflora [121], but could potentially be exploited as postbiotic ingredients in commercial sun protection products.

Furthermore, ceramides are a diverse class of lipids that make up most of the human cutaneous barrier. Up to date, close to 1000 individual ceramide species have been identified, and much remains to be discovered about their distinct roles in epidermal function [122]. With increasing age, ceramide levels diminish, and a correlation with increased trans-epidermal moisture loss and skin disease incidence is observed. Recently, Zheng et al. found that the abundant skin commensal bacterial species *Staphylococcus epidermidis* contributes to cutaneous barrier homeostasis by secreting a sphingomyelinase (SMase) that not only assists in acquiring essential nutrients for the bacteria but also helps the host in producing protective ceramides [123]. Stimulation of ceramide production on the skin through the modulation of SMase activity via microbiome-based interventions could potentially revert skin aging processes and restore moisture levels.

In a recent clinical phase II study, oral administration of a probiotic in the form of *Bacillus subtilis* led to a significant decolonization of the gut and nares by the pathogenic *Staphylococcus aureus* without disbalancing the microbiota, thus providing a valuable alternative to more frequently non-effective antibiotic treatments associated with side effects [124]. The mechanism behind host decolonization by this pathogen was shown to be mediated via a class of secreted lipopeptides, the fengycins, by *Bacillus* species that specifically inhibit *Staphylococcus aureus* quorum sensing [125]. This finding raises the hypothesis that the same concept could be applied to skin for the treatment of *Staphylococcus aureus*-associated skin infections, such as cellulitis, furuncles, and abscesses. Indeed, a proof of concept was provided by a small clinical study led by Nakatsuji and colleagues [126]. Topical application of selected probiotic *Staphylococcus epidermidis* and *Staphylococcus hominis* strains capable of producing antimicrobial, strain-specific, highly potent, and selective peptides onto the skin of human subjects with AD decreased the absolute abundance of *Staphylococcus aureus*. 

According to the Federal Food, Drug, and Cosmetic Act (FD&C Act), cosmetics are defined as articles intended to be applied to the human body for cleansing, beautifying, promoting attractiveness, or altering the appearance. While cosmetic ingredients can produce physical and bioactive effects on the skin, claims are prohibited that imply the product has a pharmaceutical effect or can treat or prevent diseases. As such, pre-, pro-, and postbiotics are required to be correctly categorized either as cosmetical, pharmaceutical, or cosmeceutical ingredients—a new category of products placed between cosmetics and pharmaceuticals that is “purported to have therapeutic action capable of affecting the skin positively beyond the time of its application” [127] and is currently not recognized by the FD&C Act. The recent literature about the evolving field of microbiome-targeting skincare products and regulatory aspects is available [113,128]. Special attention is to be given to probiotics as live biotherapeutic products with the aim to prevent or treat a disease through targeted modulation of the microbiome and a defined regulatory framework is required to include microbe-based medicines [129]. 

The skin microflora has now widely been accepted not only as a part of extrinsic factors but also to play an integral part in maintaining skin homeostasis. It will be crucial to understand how different members of cutaneous microbial communities interact with each other and the host’s skin to develop strategies for microbiome-based interventions for cosmetic applications as well as the prevention and treatment of skin disorders.

## 6. Discussion and Future Directions

During the past two decades, a lot of effort was put into understanding how the intestinal microbiota contributes to human metabolic health. Current cause-and-effect studies aim at advancing the knowledge of how gut microbiome-derived compounds are linked to the pathogenesis of metabolic diseases [130]. The skin microbiome research field has long been overshadowed by more advanced gut microbiome-oriented findings. This is partly due to a lack of high-resolution genomic data available on skin microbial species and strains, as technical limitations prevent whole-metagenome sequencing of samples with low microbial biomass. Nevertheless, interest in skin microbiome–host interactions and their implications for human health is steadily catching up.

The promising opportunities for the in situ microbial production of beneficial metabolites for skin cosmetic and therapeutic applications are undoubtedly multifaceted. In contrast to the integration of ready-to-use bioactive ingredients into skincare products, a topical application of the respective substrates allows the enzymatic in situ activation of bioactives directly at the intended location. The biochemical reaction steps are expected to be catalyzed with an efficiency that is proportional to the respective abundance of microbial activity, thereby significantly extending the shelf-life of the active compound and reducing excess dosage and associated side effects. Other potential advantages are that the substrates may be derived from a sustainable, up-cycled source, thereby reducing costs and environmental burdens.

This review provides a comprehensive state-of-the-art literature summary supporting the existence of mechanisms involved in substrate-driven skin microbial production of bioactive compounds, such as lipid mediators. Nevertheless, an extensive in vivo clinical efficacy study requiring longitudinal topical application of an n-3 PUFA solution, followed by targeted lipidomic analysis for detection and quantification of lipid mediator species on the skin, will be needed to provide ultimate support for such claims. 

In the past, research was limited to characterizing the complex compositions of skin microbial communities at a genomic level due to technical restraints. Integrative multi-omics analyses followed by accurate statistical and bioinformatic data interpretation will be required to identify and characterize biosynthetic pathways associated with specific skin microbial species. This will allow us to precisely manipulate the skin microflora composition and ultimately to topically supplement substrates for the in situ production of beneficial compounds. Alternatively, once defined, microbial pathways associated with the production of harmful metabolites can be repressed, thus improving cutaneous homeostasis (see Figure 1 and Figure 2).

Skin microbes are highly adapted to their niches within different skin regions, which reflects a longstanding relationship with their human and animal hosts. It is now evident that skin microbes not only interact with proximate cellular microenvironments but also cross-talk with other organ systems, thereby impacting lung and gut inflammation, TRPV1-mediated pain perception, and melanoma tumor growth [131]. A metabolite-mediated interaction between skin-resident microbes and distal organs would require that microbially produced metabolites penetrate the *stratum corneum* barrier to reach the circulatory system, through which they can reach other tissues and organs to unfold their effect. Similarly, as described above, targeted modulation of skin microbial metabolite production could influence distal organ homeostasis or yield neurocosmetic products that directly impact the nervous system. 

In closing, this review intends to spark interest in making use of the power of a live cutaneous microflora as a novel tunable dimension for the innovation of microbiome-based functional skincare solutions.

## Figures and Tables

**Figure 2 microorganisms-12-01711-f002:**
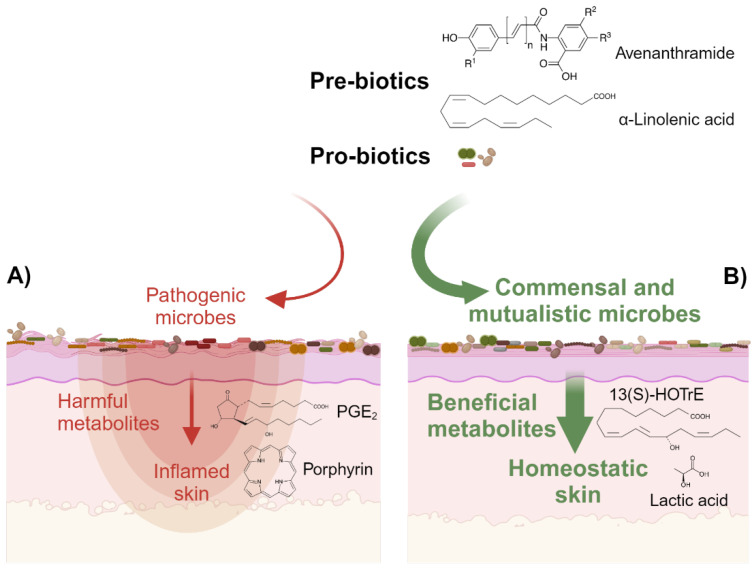
Targeted manipulation of the skin microbiome to improve skin health. Skin microbiome-derived metabolites with bioactive properties can adversely or beneficially impact skin homeostasis. (**A**) For example, acneic *Cutibacterium acnes* strains produce metabolites, such as porphyrins (structure shown), which activate the cutaneous inflammasome [81]. Moreover, acne-affected skin is locally enriched with pro-inflammatory PGE_2_ [82]. Topical application of specific probiotics, e.g., non-pathogenic *Cutibacterium acnes* strains that compete for the same niche, can indirectly reduce the production of pro-inflammatory porphyrins through the displacement of acneic *Cutibacterium acnes* strains. (**B**) The skin homeostatic equilibrium can also be supported through the production of beneficial metabolites (symbolized by thicker green arrow), for example by topical supplementation of specific prebiotics, such as Avenanthramide (structure shown) contained in colloidal oats, which supports the growth and microbial activity of mutualistic or commensal microbes, such as *Staphylococcus epidermidis* bacteria, thus enhancing the microbial production of lactic acid (structure shown) [83] with several attributed favorable effects on skin homeostasis.

## Data Availability

No new data were created or analyzed in this study. Data sharing is not applicable to this article.

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
