# Peer review of "A New Generation of Postbiotics for Skin and Scalp: In Situ Production of Lipid Metabolites by Malassezia"

_microorganisms, 2024, doi:10.3390/microorganisms12081711_

Round 1

Reviewer 1 Report

Comments and Suggestions for Authors

It is a well written review that contains a lot of informative data.

It is beneficial for the reader.

I have some questions and comments.

Several antifungal shampoos to control Malassezia are already available, what do you think?

I would like to suggest that you cite the paper by Naoya Umemoto et al, which describes the bacterial and fungal microbiome of the skin in atopic dermatitis.

Microorganisms 2024 Jan 22;12(1):224.

Dupilumab Alters Both the Bacterial and Fungal Skin Microbiomes of Patients with Atopic Dermatitis

Naoka Umemoto M Kakurai T Matsumoto ,  K Mizuno  , O Cho T Sugita T Demitsu 

Reviewer 2 Report

Comments and Suggestions for Authors

The manuscript provides an overview of existing research and available data on the effects of different Malassezia mediators in various skin conditions, as well as their importance in the interplay of the skin and its microbiome. Additionally, models are assumed that explain the potential beneficial effect of Malassezia-derived lipid mediators on skin and scalp health. However, due to the undoubted complexity and numerous unknowns, it would be good to further clarify and/or emphasize certain aspects in accordance with the comments.

Comment #1

In the introductory part, please introduce and define the terms pre-, pro- and postbiotics.

Comment #2

Page 3, line 115-118 Describe in more detail the impact of the mentioned factors on the skin microbiome.

Comment #3

Page 4 In this part, both the positive and negative effects of Malassezia are listed. How the authors consider the possibility of achieving only positive without negative effects?

Comment #4

Similar to the previous comment in the mechanism in Figure 1 only the production of positive mediators is assumed, what about the negative mediators mentioned on page 4 (167-170, 195-197) that are involved in the development and progression of skin dysbiosis.

Comment #5

Page 6, line 252-255 This is an assumption. Is there any evidence for it?

Comment #6

Page 7, line 273-276 Are there any studies to support this statement? If there is any study about positive effect of n-3 PUFA from cosmetic product on skin mention the results if not also highlite that there are no in vitro or in vivo studies to support the assumption.

Comment #7 page 7, line 311-314

I believe that it is precisely for this reason that the considerations related to the previous comments are important. Emphasize that perhaps the biggest challenge is precisely the question of which properties will prevail.

Comment #8 Page 8, line 334-341

All the discoveries related to the skin microbiome approximately 5 years ago started the trend of so-called microbiome friendly cosmetics, but the fact is that this is no longer the case and there are very good reasons for that. First of all, the manipulation of the skin microbiome can have an impact on the health of the skin, both positive and negative. Not only does insufficient knowledge of the relationship between the microbiome of the skin and the health of the skin and the whole organism call for caution when it comes to external manipulation of the microbiome, but it is also legally inadmissible for a cosmetic product to primarily affect health. That is the role of medical products, not cosmetic products. This should also be a comment related to the existence of cosmetic products that are discussed on the market. There are also papers on this within the cosmetic community.

Comment #9

Page 8 and 9 line 348-353

Is there a risk of depigmentation disorder of the surrounding normal skin that is normally associated with malassezin.

Comment #10

In view of the comments the title should be adjusted to be more in line with the currently proven and real possibilities of postbiotic Malassezia for skin and scalp.
